# Usability Testing of a Mobile Health Application for Self-Management of Serious Mental Illness in a Norwegian Community Mental Health Setting

**DOI:** 10.3390/ijerph18168667

**Published:** 2021-08-17

**Authors:** Marianne Storm, Hilde Marie Hunsbedt Fjellså, Jorunn N. Skjærpe, Amanda L. Myers, Stephen J. Bartels, Karen L. Fortuna

**Affiliations:** 1Department of Public Health, Faculty of Health Sciences, University of Stavanger, 4021 Stavanger, Norway; hildemarie.fjellsa@uis.no (H.M.H.F.); jorunn.n.skjerpe@uis.no (J.N.S.); 2School of Public Health, Rivier University, Nashua, NH 03060, USA; amyers1@rivier.edu; 3Mongan Institute, Massachusetts General Hospital, Harvard Medical School, Harvard University, Boston, MA 02114, USA; SJBARTELS@mgh.harvard.edu; 4Department of Psychiatry, Geisel School of Medicine, Dartmouth College, Lebanon, NH 03766, USA; Karen.L.Fortuna@Dartmouth.edu

**Keywords:** mobile health, user-testing, usability study, serious mental illness, peer support

## Abstract

Background: For digital tools to have high usability and fit service users’ health needs and socio-environmental context, it is important to explore usability with end-users and identify facilitators and barriers to uptake. Objective: To conduct user testing of the smartphone health application, PeerTECH, in a Norwegian community mental health setting. Methods: Semistructured interviews and usability testing of the PeerTECH app using the Think-Aloud approach and task analysis among 11 people (three individuals with a serious mental illness, two peer support workers, and six mental health professionals). Results: Study participants perceived PeerTECH as a relevant tool to support self-management of their mental and physical health conditions, and they provided valuable feedback on existing features as well as suggestions for adaptions to the Norwegian context. The task analysis revealed that PeerTECH is easy to manage for service users and peer support workers. Conclusions: Adapting the PeerTECH smartphone app to the Norwegian context may be a viable and useful tool to support individuals with serious mental illness.

## 1. Background

Mobile health applications are being developed at a fast pace and show promising evidence of promoting positive health behavior change such as monitoring physical activity and mental health symptoms [1,2]. Individuals with a serious mental illness (i.e., schizophrenia spectrum disorders, major depressive disorders and bipolar disorders, personality disorders, along with persistent functional impairment [3]), could benefit from using digital tools such as mobile health applications (apps) that promote health literacy and positive behavior changes [1,2,4], as they experience early mortality health disparity and die up to 32 years earlier than the general population, mainly due to poor health behaviors [5,6]. As such, technology-supported integrated medical and psychiatric self-management interventions for people with serious mental illness have received attention. One example is the PeerTECH smartphone health app (PeerTECH app) developed by Digital Peer Support in Nashua, NH, USA. The term PeerTECH originates from the integration of technology into traditional peer support services [7,8,9]. PeerTECH is informed by Integrated Illness Management and Recovery (I-IMR) for individuals with a serious mental illness and medical comorbidity and helps to self-manage medical and psychiatric conditions [8]. A core feature of the PeerTECH app is the components of peer support, sharing lived experience, and self-determination. PeerTECH is delivered by a peer support specialist and is designed to improve self-management and empower people to address the vulnerability that may lead to new or worsening medical and psychiatric symptoms and conditions. Key components in the PeerTECH app include the following: (1) education to improve knowledge about managing mental and medical illness (i.e., psychoeducation, medical management, social skills training); (2) healthy behaviors (e.g., diet, exercise) and behavior change; (3) training and planning to prevent relapses, and (4) increased coping skills from defining personalized goals associated with medical and mental health, learning from similar others lived experiences of recovery and referral to resources, as needed [10,11].

For a digital tool, such as the PeerTECH app, to have high usability and likelihood to fit the context and end users’ mental and physical health needs, it is important to identify the use of existing digital tools in service provision, as well as facilitators and barriers to uptake and use of the mobile app in various contexts [2,12]. Usability can be defined as the “*extent to which a system, product or service can be used by specified users to achieve specified goals with effectiveness, efficiency, and satisfaction in a specified context of use*” [13]. Usability is a factor in the adoption of mobile health apps when people who need to use them may have problems interacting with an app due to their health conditions [2,7]. For example, usability testing provides a unique opportunity to promote engagement and have direct data on the end users’ interactions with the technology, thus providing a picture of how useful the system will be for individuals with serious mental illness within their socio-cultural environment [14]. A recent systematic review by Borghouts et al. identified social connectedness, increased insight into one’s own health, and a feeling of being in control, as important factors facilitating continuous user engagement with a digital mental health intervention [4].

We conducted usability testing of the American prototype PeerTECH app to test PeerTECH with Norwegian end-users (i.e., mental health service users, peers support workers) and mental healthcare professionals. Early inclusion of relevant end-users in the development of a Norwegian version of PeerTECH is important to get feedback on what is important to end-users, as well as needs for adjustments in the content and the design of the smartphone app to adapt it for use with individuals with serious mental illness receiving community mental health services [12,15]. Community engagement and user participation when developing and adapting digital interventions can ensure relevance and appropriateness [16] and facilitate continuous engagement with health interventions [4]. The objective of this study is to assess the usability of the PeerTECH app with relevant end-users engaged in Norwegian community mental health services, and to identify facilitators and barriers to uptake and implementation of PeerTECH. In this paper, we use the term community mental health services when referring to services provided to residents with a serious mental illness in the municipality.

## 2. Methods

### 2.1. Study Design

We performed usability testing of the American prototype PeerTECH app with a purposive sample [17] of mental health service users, peers support workers and mental health professionals from community mental health services in two municipalities in Western Norway. The usability testing included individual semistructured interviews to explore facilitators and barriers to the use of digital tools in community mental health services and perceived utility of the PeerTECH app, “Think-Aloud” method, and task analysis for direct testing of key aspects of the usability of PeerTECH app. The “Think-Aloud” method allowed end-users to say aloud their thoughts, feelings, and the observations they made when testing the smartphone application [7,14,18]. Task analyses were employed to assess features and the user interface of the PeerTECH app [18] with service users and peer support workers.

### 2.2. Study Setting, Recruitment, and Participants

Norway has a publicly-funded health service system. Municipalities are responsible for providing health services (primary, mental health) and social services to their residents [19]. The municipalities offer mental health services based on assessments of individual resources, self-management capacity, and primary care needs. The services include preventive measures, training and supervision, rehabilitation, and assistance to support the individual’s independent living and meaningful life in the community [20]. Study participants were recruited from two similar-sized municipalities in the western part of Norway with a population of approximately 20,000 inhabitants in each municipality.

We contacted the community mental health services via email with information about the research project to assess interest in participating in the study. In the first municipality, two mental health professionals were interested and agreed to participate in an interview. From the second municipality, the researchers were invited to a staff meeting to inform the team about the project, the PeerTECH app, and to explore interest in participating in usability testing. Interviews were then scheduled and conducted with service users, peer support workers, and mental health professionals based on the participants’ interest and experience with using digital tools and smartphone apps and willingness to participate in the study. Data collection was conducted by the first author and was carried out during the daytime in the offices of the community mental health services. Each interview included “Think-Aloud” and task analyses, and lasted between 30 min to 1.5 h.

Participants were provided written information about the project, including details regarding voluntary participation and how confidentiality would be ensured. All subjects gave their informed consent before participating. The project was approved by NSD—Norwegian center for research data (id 769409) and was conducted according to the guidelines of the Declaration of Helsinki.

There was a total of 11 study participants including three individuals with serious mental illness (one woman, two men, ages ranging between 20–60 years) receiving community mental health services at home, two peer support workers (one woman and one man, ages ranging between 20–50 years), and six mental health professionals specialized in mental health nursing and social work, (six women, ages ranging between 20–50 years). All the study participants owned a smartphone and reported using it daily.

### 2.3. PeerTECH App

The PeerTECH app is designed for service users to reinforce skills learned from in-person sessions with a peer support specialist. PeerTECH includes the following features: (1) a peer support specialist facing app with a curriculum to guide the peer support specialist during in-person meetings with the service user to deliver and support self-management skill development; (2) a patient-facing app that includes: (a) access to personalized self-management support; (b) the service users’ personal wellness needs and goals; (c) a chat feature for use between peer support specialist or a mental health professional and the service users’ smartphone app to support self-management gains; and (d) an on-demand library including self-management videos and materials to discuss the interconnection between mental health, physical health, and social health, the role of stress in mental and physical health, coping skills training, and lived experiences of self-management challenges and successes. The PeerTECH library includes classes to be conducted by the service user together with a peer support specialist during weekly in person-meetings or using the phone. Additionally, library features can be accessed offline.

As illustrated in Figure 1, The PeerTECH app can be accessed by service users and peer support specialists from either a mobile phone or a tablet. The service user interface includes access to messaging directly with an assigned peer support specialist, goals, wellness, surveys, and a resource library on the home screen. The peer support interface allows peer support specialists to message directly with assigned service users, view service users’ goals and wellness plans, and view service users’ progress through the resource library.

### 2.4. Data Collection

We used a semistructured interview guide based on the Consolidated Framework for Implementation Research (CFIR), which is a practical guide for assessing barriers and facilitators when preparing for the implementation of an innovation such as PeerTECH [21]. The guide included four open-ended questions about the use of mobile smartphone apps and digital technologies in community mental health services, facilitators, and barriers to the use of digital tools in community mental health services. We used the same interview guide with mental health service users, peer support workers, and mental health professionals.

We applied the “Think-Aloud” method and task analysis stepwise with the researcher taking notes for each step. In Step 1 the participants were asked to say aloud their first impression of the PeerTECH app. We asked about: the perceived usefulness and relevance of the PeerTECH app to their situation, and suggestions for elements that could be added to adapt the PeerTECH app to the Norwegian context for mental health service provision in the municipality. In Step 2 we conducted task analysis with service users and peer support workers including scoring of task performance on a scale from 0–3 (0 = cannot do the assignment, 1 = carry out the assignment with physical guidance, 2 = carry out the assignment with oral guidance, 3 = carry out the assignment independently). We asked participants to give their oral feedback when performing the following specific tasks: (1) open the app, (2) start one of the learning modules in the app, (3) start and stop the video, (4) assess the volume, (5) open the icon goals (6) write a proposal for a personal goal, (7) open the icon Messages, and (8) change the text size.

### 2.5. Data Analysis

Written notes were taken by the researcher during the individual interviews when the participants were “Thinking-Aloud”, and while conducting task analysis as participants interacted with the PeerTECH app, and constitute the qualitative data material. Data analysis was informed by the study objective to assess the usability of the PeerTECH app for the Norwegian community mental health services and facilitators and barriers to uptake of PeerTECH. We were interested in the study participants’ views on the information provided within the PeerTECH app (i.e., too much or too little information and information content), perceptions of graphics and text, difficulties with navigating, overall understanding of the PeerTECH app, and perceived relevance for Norwegian end-users. These aspects have been reported to be relevant aspects when assessing feasibility studies of information systems [22]. The task analysis included numeric scorings of how well the participant performed the task assignment.

## 3. Results

### 3.1. Digital Technology Tools for Self-Management in Mental Health Service Provision

According to the interview participants, there are few digital tools used in community mental services in the municipalities. Mental health professionals had some experience with a computer-based individualized care planning system containing the service users’ personal goals and measures to facilitate collaboration between the service users, family members, mental health professionals. However, the professionals mentioned barriers to the use of the care planning system including a poor user interface, a complicated login procedure, insufficient applicability for mobile devices, and employee resistance.

Assisted self-help for individuals with anxiety disorders and moderate depression, includes a psychology-based computer program including phone follow-up from community mental health care to assess symptom reduction. The program contains learning modules about anxiety and depression and how to cope, as well as quizzes and movie clips. According to the interviewees who had used the assisted self-help program with service users, it had been fun to use and the program was flexible and easy to use. However, the program did not include peer support, physical and dental health, and how to get the medical and mental health help needed.

Interviewees also mentioned a Norwegian smartphone app that could be downloaded in the App store called “My crisis plan”. The app is a tool where the service user can register warning signs, strategies to cope with increasing symptoms, a social network, mental health helpline, an emergency helpline for children and adolescents, as well as a map and search function to find the nearest emergency room. One mental health professional used the app with a service user and reported that it was useful for the service user to access his/her crisis plan when in the emergency room and being able to share the plan with the staff.

### 3.2. Communication via Text Messaging

Participants were positive about sending text messages in PeerTECH. They said that communicating via text messaging using mobile phones is a common and easy way to communicate between professionals, peer support workers, and service users. A few service users also used Snapchat for their communication with professionals. According to mental health professionals, some service users preferred sending short texts about how they feel and what they are struggling with. Service users considered texting with professionals as easy; it provides a distance, and at the same time it assures that the message has been read and that it will open for feedback or a specific focus in the next conversation. Professionals also considered text messages as useful starting points for their upcoming meetings and conversations with service users and commonly respond to the messages with “*I see you have sent a message—I will read it and get in touch with you again*”. A peer support worker said “*It is less scary to write than to speak. It is easier to write down, it will be your own words and at the same time it gives a focus to the conversation and ensures that you get feedback on what you want*”. Texting is an opportunity for not just talking but is a tool to support the service users’ self-management of mental illness.

### 3.3. Graphic Design of PeerTECH Home Screen

Interview participants commented that the graphic design of the home screen of the PeerTECH app (Figure 2a) was “*nice*” and “*understandable*”. They praised the pictures and reported that the app was easy to navigate. The first module page in the app includes five icons—survey, wellness plan, goals, library, and messages, and was perceived to be informative, simple, and provided a good overview of the content of the app. A few participants suggested adding an icon called resources. Participants also commented on the white background and suggested including photos or a more colorful background. One participant suggested that the app could have had a slide function, meaning that the various icons could be slid to be rearranged (i.e., first goals, then resources, then wellness plan).

### 3.4. Perceived Relevance of Goals in the PeerTECH App

Goals were perceived as useful to participants. The service users and peer support workers related goals to their situation, and some users provided examples of their own goals, that they had many types of goals, how these goals had changed over time and that they had achieved their goals and set new goals. One service user said “*To begin with, my goal was to remember to take my meds every night, then my goal was to remember to take my meds morning and evening. Eventually, I had a goal to start school. Soon, I have completed this goal and received a bachelor’s degree*”. Participants suggested *Goals* include short instructions or a class about how to develop their own goals, provide a few examples of goals, and mark goals achieved to visualize progression and for positive reinforcement.

### 3.5. Perceptions of the Content of Information in PeerTECH Library

The PeerTECH app library contains 14 modules. Each module contains goals including instructions, discussion points with a peer, a short film, tips for how to create your wellness plan, follow-up on goals, and suggestions for home practice and feedback. Feedback on the *library* was that the modules were informative and relevant. Regarding the length of the modules, one participant commented, “*I think they should not be longer than 5–10 min. These are my courses, and each course must be able to be done on the bus or at the doctor’s office or when I wait for class.*”

Participants considered the module on oral health as particularly important and that specific information about rights within the Norwegian health system to oral treatment for individuals with a serious mental illness could be added. One professional said, “*If the service user has visits from mental health services once a week over three months, they have a legal right to free oral treatment*.”

We tested the PeerTECH library module “*Recovery as a daily process*” (Figure 2b) which was perceived as highly relevant across service users, peer support workers, and professionals. Service users related the specific content and what to do with their own situations and warning signs. One peer support worker said that her warning signs were “*little sleep, little food, suicidal thoughts, high music. If these symptoms occur, I am to talk to my boyfriend, my mom, and dad, or my general practitioner.*” Participants valued the opportunity to put their warning signs and measures in the wellness plan. Another comment received was that the concept of the wellness plan resonated well with both recovery and health.

### 3.6. Task Analysis of Navigating in PeerTECH App

Five participants (service users and peer support workers) performed eight task assignments to assess their ability to navigate in the PeerTECH app. The task assignments were performed on an iPhone used by all participants. Each task assignment was scored by the interviewer using the codes: 3 = carry out the assignment independently; 2 = Carry out the assignment with oral guidance; 1 = Carry out the assignment with physical guidance; 0 = Cannot do the assignment. The results from the task analysis with service users (*n* = 3) and peer support workers (*n* = 2) are presented in Figure 3 below.

The three task assignments: Enter the library course “Recovery as a daily process;” Go through the library course “Recovery as a daily process;” and “Write a proposal for a goal,” were performed independently by all participants. For the task assignments, start and stop video about “Recovery as a daily process,” navigate to the icon “goal,” and navigate to the icon “messages,” one participant needed verbal guidance. For the two task assignments, check the audio level of video playback and try to change the text size, several participants required physical guidance to carry out the task. Four participants were not able to change text size. The tasks were performed on an iPhone, and the participants explained the difficulties with checking the audio level and changing the text size due to their lack of familiarity with the iPhone.

## 4. Discussion

The objective of this study was to assess the usability of the American prototype PeerTECH app with relevant end-users in Norwegian community mental health services and to identify facilitators and barriers to the uptake of PeerTECH. Study results showed that there are some digital tools available to service users. The “Think-Aloud” method enabled the study participants to relate the PeerTECH app to their situations. Participants perceived the app to be useful for texting and a tool to support self-management of their health condition, and they provided valuable feedback on existing features of the app as well as suggestions for adaptions that could facilitate uptake and use. The task analysis revealed that the PeerTECH app is easy to manage for both service users and peer support workers, which are the key end-users.

Our study presents novel results from usability testing of the prototype PeerTECH app with Norwegian end-users, including a patient-facing app that offers access to personalized self-management support; the service users’ personal wellness needs and goals; a chat feature for text-messaging between peer support specialist or a mental health professional and the service users’ smartphone app to support self-management gains; and an on-demand library of self-management resources. Similar usability testing with the prototype PeerTECH app has not been performed with American end-users. Our study included usability testing combined with individual interviews with service users, peer support workers, and mental health professionals in community mental health services. Future usability testing could also include other types of direct testing such as “Question Asking Method” and “Task performance Measurement” and physiological monitoring technology (i.e., blood pressure, heart rate, head, and eye-tracking), as well as testing usability in both simulated and real-time environments [14]. For future digital technologies, the mobile app development phase can include a mix of data collection methods (questionnaire survey, observations, focus-group interviews, self-reporting logs, workshops) and can be an interactive process over time [23]. An iterative approach is deemed necessary when smartphone apps are to be a key component in a health intervention in which the result of usability testing informs choices and decisions made relating to study design and content [12]. Although our study sample of service users (*n* = 3) and peer support workers (*n* = 2) was small, we believe they were representative of those who are expected to use the app in Norway and were aligned with usability testing requirements [22]. We also included the perspectives of mental health professionals to get information about available digital tools and inform the content of the Norwegian version of the PeerTECH app, which is a strength. Mental health professionals have not been included in pilot studies of the PeerTECH app in the US [7,8]. We are aware that the PeerTECH app does not include communication between the service users and family members or friends which could be useful. Our study also did not consider the perspective of caregivers, family, and friends, which can play a vital role in the recovery process for people with serious mental illness.

Available digital tools mentioned by the study participants were a computer-based individualized care planning system, a psychology-based computer program “Assisted self-help” for individuals with anxiety disorders and moderate depression, and an app “My crisis plan” to register warning signs, strategies to cope with symptoms, social network, also including a mental health emergency helpline and a map and search function to find the nearest emergency room. None of the available digital tools or interventions mentioned by the study participants were peer-led, targeted health literacy, physical and mental health, well-being, and positive behavior change in individuals with a serious mental illness. A recent systematic literature review performed by Fortuna and colleagues [16] identified 11 studies that implemented peer-delivered interventions supported with technology to individuals with a serious mental illness. The interventions targeted shared decision-making, cognitive therapy, physical well-being, and weight management. Peers used smartphone apps, text messaging, web-based platforms and fitness tracking for interaction with service users and delivering services. Only one study conducted by Fortuna et al. [9] targeted integrated medical and psychiatric self-management and combined in-person visits with a smartphone app. Promising evidence from our pilot studies indicates that PeerTECH is feasible, acceptable, engaging, safe, effective, and has been delivered with high fidelity [8,9,24]. Our results from usability testing can be considered a valuable first step in adapting the PeerTECH app to the needs of Norwegian service users and peer support workers. The next steps will require translating the text and video content, as well as adding context-specific content to tailor the app for use in a peer-led health and well-being intervention. Such an intervention can be tested in a randomized-controlled trial to assess the effectiveness of PeerTECH on health literacy, well-being, behavior change, and service integration for service users with a serious mental illness. It can also be useful to assess the potential for implementation and use of PeerTECH in Norwegian community mental health services.

## 5. Conclusions

Engaging service users, peer support workers, and mental health professionals in considering what is important to them will enable us to ensure the contextual fit of the PeerTECH app to service users’ needs. This is particularly important for a vulnerable population, such as individuals with serious mental illness, who bear a high burden of diseases and potential cognitive limitations that may impair the ability to engage in self-management for both physical and mental health. The study provided relevant knowledge about the use of existing digital tools in community mental health services and factors that may influence the use of the PeerTECH app.

## Figures and Tables

**Figure 1 ijerph-18-08667-f001:**
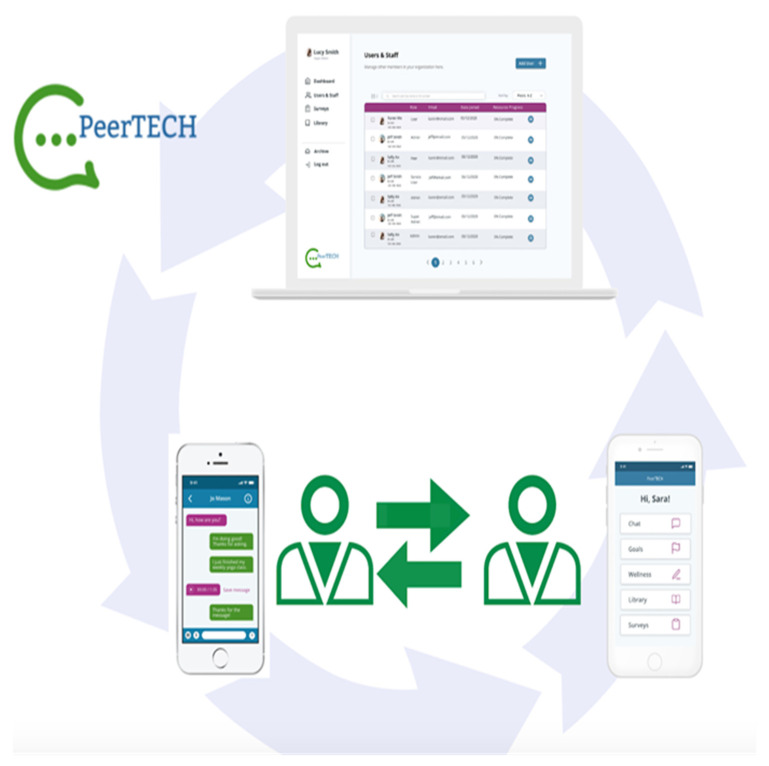
Illustrates the PeerTECH mobile app.

**Figure 2 ijerph-18-08667-f002:**
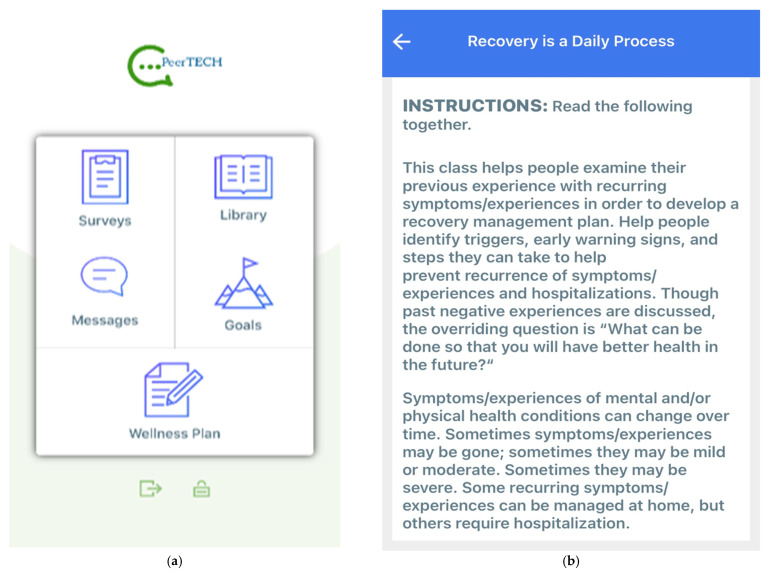
(**a**). Screenshot of the home screen in PeerTECH. (**b**). Screenshot of the instructions to the library module “Recovery is a Daily Process”.

**Figure 3 ijerph-18-08667-f003:**
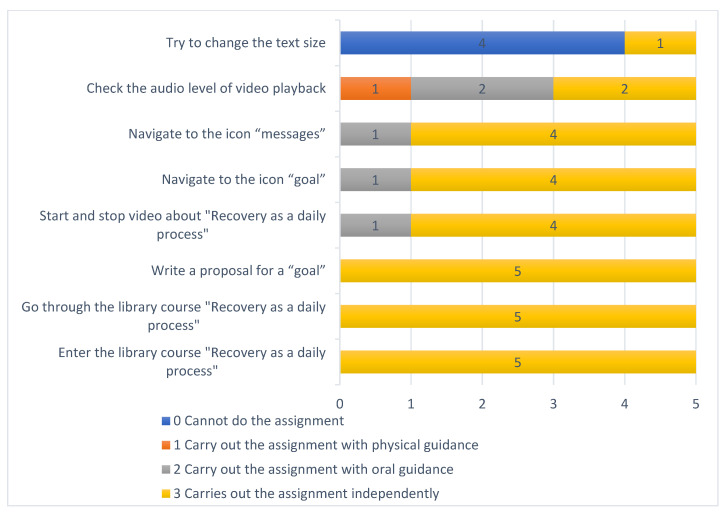
Task analysis of navigating the PeerTECH app.

## Data Availability

The Norwegian dataset generated for this study is available on request to the first and corresponding author.

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
