# Peer review of "Usability Testing of a Mobile Health Application for Self-Management of Serious Mental Illness in a Norwegian Community Mental Health Setting"

_ijerph, 2021, doi:10.3390/ijerph18168667_

Round 1
Reviewer 1 Report
A usability testing was conducted on a smartphone health application (PeerTECH; american prototype) in the context of mental health services in Norway using semi-structured interviews. The authors conclude that this app may be useful to support people with serious mental illness.
Major Points
- This important conclusion is not supported by the data; the sample size is far too small (only three end-users) and the three individuals with a serious mental illnesses were not well characterized (diagnosis according to DSM5 criteria was not specified). A much larger study with better characterized study populations is required. My recommendation is to study the three categories (people with mental illness, peer support workers, and mental health professionals) separately with a minimum of 20 participants/category.
- The study does not take into account another important category that plays a vital role in the rehab process for people with serious mental illness: caregivers as well as friends/family members.
- Given that the PeerTECH does not provide any practical management support, it is odd that study participants perceived this app as a relevant tool to support self-management of their mental and physical health conditions. Small sample size may have shifted perception in one direction (i.e. positive perception).
Minor Points
- Background: Please condense the three terms schizophrenia, schizoaffective and psychotic disorder under the term schizophrenia spectrum disorders.
- Study setting, recruitment and participants: The authors state they contacted a number of the community mental health services regarding the research project to check if there was interest to participate in the study. From the two municipalities contacted a total of 8 peer support workers and mental health professionals agreed to participate. In order to understand the potential appeal of the app under investigation it is important to state the total number of people invited and to express as percentage the number of people who agreed to participate. Please specify the specialty for the six mental health professionals.
- PeerTECH app: Please describes the theoretical background at the basis of the App. Please explain in details its functioning, add more figures about its functioning and explain how is similar or innovative compared with other similar App.
- Results: Assisted self-help for individuals with anxiety disorders and moderate depression, is a psychology-based computer program including phone follow-up from community mental health care to assess symptom reduction. This is description but it a refer to common and not serious mental illness such ad anxiety and moderate depression.
- Figures: Fig. 1 cannot be read (too small) and it is not helpful. Fig. 2 legend requires a better description of the app flow.
Author Response
Comments Reviewer 1:
A usability testing was conducted on a smartphone health application (PeerTECH; american prototype) in the context of mental health services in Norway using semi-structured interviews. The authors conclude that this app may be useful to support people with serious mental illness.
Major Points
This important conclusion is not supported by the data; the sample size is far too small (only three end-users) and the three individuals with a serious mental illnesses were not well characterized (diagnosis according to DSM5 criteria was not specified). A much larger study with better characterized study populations is required. My recommendation is to study the three categories (people with mental illness, peer support workers, and mental health professionals) separately with a minimum of 20 participants/category.
Response to the reviewer: Thank you for this important comment. We are aware of the small sample size of service users and peer support workers and have addressed this in the discussion about study limitations. However, the study assesses the usability of the PeerTECH app with end-users (service users, peer support workers) and is consistent with usability studies. According to Kushniruk & Patel (2004) a considerable amount of information can be obtained from a small number of subjects when they are representative of target users of the system being assessed. We also included mental health professionals to have their perspectives on the relevance of the app. This study has provided important data valuable in assessing the actual usability of PeerTECH within the Norwegian context. Usability is important to develop a product that is usable by a population and it is also an important first step on the software development lifecycle.
The study does not take into account another important category that plays a vital role in the rehab process for people with serious mental illness: caregivers as well as friends/family members.
Response to the reviewer: Thank you for this important comment. We have addressed the limitation of our study not including caregivers, family members, and friends. However, please note the PeerTECH app is designed to empower peer support specialists to deliver an intervention independently.
Given that PeerTECH does not provide any practical management support, it is odd that study participants perceived this app as a relevant tool to support self-management of their mental and physical health conditions. Small sample size may have shifted perception in one direction (i.e. positive perception).
Response to the reviewer: Please note that PeerTECH provides practical management support as it offers text and videos on practical support. See our previous comment about sample size in usability studies.
We have included text in the methods section to better explain how PeerTECH functions. PeerTECH includes the features: (1) a peer support specialist facing app (on a smartphone) with a curriculum to guide peer support specialists during in-person meetings to deliver self-management skill development. (2) a patient-facing app that offers self-management support, an on-demand library of self-management resources; and (3) text messaging between peer support specialists and service users between meetings to support self-management gains.
Minor Points
Background: Please condense the three terms schizophrenia, schizoaffective and psychotic disorder under the term schizophrenia spectrum disorders.
Response to the reviewer: We have condensed the three terms as suggested in the introduction.
Study setting, recruitment and participants: The authors state they contacted a number of the community mental health services regarding the research project to check if there was interest to participate in the study. From the two municipalities contacted a total of 8 peer support workers and mental health professionals agreed to participate. In order to understand the potential appeal of the app under investigation it is important to state the total number of people invited and to express as percentage the number of people who agreed to participate. Please specify the specialty for the six mental health professionals.
Response to the reviewer: Thank you for this important comment. All the participants that were invited to participate agreed to participate. Due to that this is a feasibility study our study sample is small and aligned with sample size in usability studies (see our previous comment about sample size).
We have included information about the specialty of mental health professionals in the text.
PeerTECH app: Please describes the theoretical background at the basis of the App. Please explain in details its functioning, add more figures about its functioning and explain how is similar or innovative compared with other similar App.
Response to reviewer: Thank you for this important comment. We have included text to explain the theoretical basis for PeerTECh in the introduction and how PeerTECH functions in the methods section. PeerTECH was informed by Integrated Illness Management and Recovery (I-IMR) for individuals with a serious mental illness and medical comorbidity and how to self-manage medical and psychiatric conditions [8].
We have added the following text in the methods section describing PeerTECH: The PeerTECH app can be accessed by service users and peer support specialists from either a mobile phone or a tablet. The service user interface includes access to messaging directly with an assigned peer support specialist, goals, wellness, surveys, and a resource library on the home screen. The peer support interface allows peer support specialists to message directly with assigned service users, view service users’ goals and wellness plans, and view service users’ progress through the resource library. Service users are assigned to peer support specialists through an admin dashboard which is accessible through a desktop web browser.
This apps differs from other similar apps as it is based on evidence-based interventions and best practices, was designed in partnership with peers, and its design functions were developed to align with accessibility and the unique impairments that may affect individuals with a lived experience of serious mental illness.
Results: Assisted self-help for individuals with anxiety disorders and moderate depression, is a psychology-based computer program including phone follow-up from community mental health care to assess symptom reduction. This is description but it a reference to common and not serious mental illness such ad anxiety and moderate depression.
Response to the reviewer: Thank you for this comment. We are aware that the assisted self-help program is not addressing individuals with serious mental illness. We have included in the discussion that the available digital tools in the community mental health services in the municipalities do not focus on individuals with serious mental illness, peer support, and improving their health and wellbeing.
Figures: Fig. 1 cannot be read (too small) and it is not helpful. Fig. 2 legend requires a better description of the app flow.
Response to the reviewer: Thank you for this important comment. We have uploaded a new version of figure 1 and included a better description of the app flow in the manuscript text. We have also included new versions of the photos in figure 2 and refer to these as figure 2a Screenshot of the home screen and figure 2b. Screenshot of the instructions to the library course “Recovery is a Daily Process”.
Reviewer 2 Report
The authors evaluated the usability of a mobile health application in a Norwegian community mental health setting. There are a few places which could be improved in the manuscript.
- Please proofread the manuscript carefully to correct the typos and missing capitalization of the first letter in a few sentences.
- The authors may want to introduce the full term of "PeerTECH" or the origin of the name.
- Figures' quality must be improved. There is not much info included in Fig.1. The contents on the interface could not be read. Fig.2. seems to be very loosy and unprofessional.
- Table quality must be improved. The text needs to be aligned properly in the table.
- The authors have a lot of self-citations related with the "PeerTECH" tool, some of them seem to be not very relevant. However, there seems to lack of a brief overview and introduction of the tool, especially to the first time audiences who are not familiar with the tool.
- The authors may want to add more discussions of the limitations of the tool, and proposed meaningful future work as next step.
Author Response
Comments Reviewer 2:
Please proofread the manuscript carefully to correct the typos and missing capitalization of the first letter in a few sentences.
Response to the reviewer: The manuscript has been carefully proofread.
The authors may want to introduce the full term of "PeerTECH" or the origin of the name.
Response to the reviewer: The term PeerTECH originates from the integration of technology into traditional peer support services.
Figures' quality must be improved. There is not much info included in Fig.1. The contents on the interface could not be read. Fig.2. seems to be very loosy and unprofessional.
Table quality must be improved. The text needs to be aligned properly in the table.
Response to the reviewer: Thank you for the important comments about the unclarity in the content and quality of the format of figures and tables. We have uploaded new versions of figure 1 and figure 2. Figure 2 is in the revised manuscript referred to as figure 2a Screenshot of the home screen and figure 2b. Screenshot of the instructions to the library course “Recovery is a Daily Process”. We have also changed the format of table 1: Task analysis of navigating in the PeerTech app. into a diagram for improved readability.
The authors have a lot of self-citations related with the "PeerTECH" tool, some of them seem to be not very relevant. However, there seems to lack of a brief overview and introduction of the tool, especially to the first time audiences who are not familiar with the tool.
Response to reviewer: This is an important comment. We have included text about the functioning of PeerTECH in the methods section and we explain the term PeerTECH in the introduction.
The authors may want to add more discussions of the limitations of the tool, and proposed meaningful future work as the next step.
Response to the reviewer: We have added more discussion of the limitations of PeerTECH and elaborated on future research and work as the next steps.
Reviewer 3 Report
The study aims to assess the usability of the PeerTECH app with ends-users in the Norwegian community mental health services. The testing included, individual semi-structured interviews, "Think-Aloud" method, and task analyses. 11 human subjects, including patients, support workers, and health professionals, were invited for the testing and interviews.
The results indicate that the PeerTECH app is perceived to be a useful tool for supporting self-management of mental health in the community.
This study only focuses on a single mobile app. The authors are encouraged to further elaborate the future work and how the proposed methods can be considered in the testing of other similar mental health self-management apps. A comparison between the chosen methods and other methods should also be discussed.
There are a few minor typos.
Author Response
Comments Reviewer 3:
The study aims to assess the usability of the PeerTECH app with ends-users in the Norwegian community mental health services. The testing included, individual semi-structured interviews, "Think-Aloud" method, and task analyses. 11 human subjects, including patients, support workers, and health professionals, were invited for the testing and interviews.
The results indicate that the PeerTECH app is perceived to be a useful tool for supporting self-management of mental health in the community.
Response to the reviewer: Thank you for these comments
This study only focuses on a single mobile app. The authors are encouraged to further elaborate the future work and how the proposed methods can be considered in the testing of other similar mental health self-management apps. A comparison between the chosen methods and other methods should also be discussed.
Response to reviewers: We have added more discussion of the future research and work. Usability testing is an important and necessary step in developing a product, a digital tool, that is usable by a population. This study is the first step on the software development lifecycle of a Norwegian version of PeerTECH.
There are a few minor typos.
Response to the reviewer: We have carefully proofread the article.
Reviewer 4 Report
It is not enough that the usability testing was conducted among only 11 participants.
,What's the novelty ?
Author Response
Comments Reviewer 4:
It is not enough that the usability testing was conducted among only 11 participants.
Response to reviewer: This is the first step on the software development lifecycle of PeerTECH for Norwegian end-users. Our study sample is small but in line with recommendations for usability testing to develop a product that is usable by a population. According to Kushniruk & Patel (2004), a considerable amount of information may be obtained from a small number of subjects when they are representative of target users of the system being assessed.
What's the novelty ?
Response to reviewer: The novelty with PeerTECH app is that it is based on evidence-based interventions and best practices, was designed in partnership with peers, and its design functions were developed to align with accessibility and the unique impairments that may affect individuals with a lived experience of serious mental illness.
Round 2
Reviewer 2 Report
The manuscript is improved in the revised version.
Author Response
Response to reviewer 2:
The manuscript is improved in the revised version.
Response to the reviewer: Thank you for this comment. We also thank the reviewer for the time and consideration.
Reviewer 4 Report
The PeerTECH app is the American prototype. What's the difference between the tests PeerTECH with Norwegian end-users and the test data from the American manufacture? What's the difference of the software development lifecycle of a Norwegian version of PeerTECH and other versions?
A comparison between the PeerTECH app and other similar mental health self-management apps should be discussed.
Author Response
Dear reviewer,
We thank the reviewer for the important comments.
We have incorporated our responses to the reviewer’s comments in our revised manuscript. Listed below are the original reviewers’ comments, followed by our responses. Revisions in the manuscript are in a blue font using track- changes.
Thank you for your time and consideration,
Response to reviewer comments:
The PeerTECH app is the American prototype. What's the difference between the tests PeerTECH with Norwegian end-users and the test data from the American manufacture?
Response to the reviewer: Thank you for this important comment. One difference is the peer support end-user in Norway which is different than Americans. The US peer system was developed after institutionalization, and they are part of a strong anti-psychiatry movement.
Our study included mental health professionals to get information about available digital tools in the Norwegian mental health service system, the perceived relevance of the PeerTECH app, and to inform the content of the Norwegian version of PeerTECH. Mental health professionals have not been included in the earlier test of PeerTECH in the US.
In the revised manuscript (page. 10) we have added the sentence: “Mental health professionals have not been included in pilot studies of the PeerTECH app in the US [7,8]”.
What's the difference between the software development lifecycle of a Norwegian version of PeerTECH and other versions?
Response to the reviewer: This is a very good question, and we appreciate an opportunity to include additional text in the manuscript. Of note, our study presents novel results from usability testing of the PeerTECh prototype with Norwegian end-users. Usability was not studied on the PeerTECH prototype version with American end-users. Our pilot studies with American users describe the Adapting process of a Psychosocial Intervention for Smartphone Delivery to Adults with Serious Mental Illness and descriptions of certified peer specialists integration of peer philosophy into the delivery of a self-management intervention enhanced with mobile health.
In the revised manuscript (page. 9) we have added the text: “Our study presents novel results from usability testing of the prototype PeerTECH app with Norwegian end-users including a patient-facing app that offers access to personalized self-management support; the service users’ personal wellness needs and goals; a chat feature for use between peer support specialist or a mental health professional and the service users’ smartphone app to support self-management gains; and an on-demand library of self-management resources. Similar usability testing with prototype PeerTECH app has not been performed with American end-users”.
A comparison between the PeerTECH app and other similar mental health self-management apps should be discussed.
Response to the reviewer: This is an important comment. We have added text in the discussion comparing PeerTECH with similar mental health self-management apps. We have also included a reference to a relevant systematic review: Digital Peer Support Mental Health Interventions for People With a Lived Experience of a Serious Mental Illness: Systematic Review.
In the revised manuscript (page 10) we have added the following text: “A recent systematic literature review performed by Fortuna and colleagues [24] identified 11 studies that implemented peer-delivered interventions supported with technology to individuals with a serious mental illness. The interventions targeted shared decision-making, cognitive therapy, physical well-being, and weight management. Peers used smartphone apps, text messaging, web-based platforms, and fitness tracking for interaction with service users and delivering services. Only one study conducted by Fortuna et al. [9] targeted integrated medical and psychiatric self-management and combined in-person visits with a smartphone app.”